# Human ABCB1 with an ABCB11-like degenerate nucleotide binding site maintains transport activity by avoiding nucleotide occlusion

**Katalin Goda**[1o], **Yaprak Dönmez-Cakil**[2,3o], **Szabolcs Tarapcsák**[1,4], **Gábor Szalóki**[1], **Dániel Szöllősi**[2], **Zahida Parveen**[5,6], **Dóra Türk**[7], **Gergely Szakács**[7,8], **Peter Chiba**[5]*, **Thomas Stockner**[2]*

1 Department of Biophysics and Cell Biology, Faculty of Medicine, University of Debrecen, Egyetem tér, Debrecen, Hungary, 2 Institute of Pharmacology, Center for Physiology and Pharmacology, Medical University of Vienna, Waehringerstrasse, Vienna, Austria, 3 Department of Histology and Embryology, Faculty of Medicine, Maltepe University, Maltepe, Istanbul, Turkey, 4 Doctoral School of Molecular Cell and Immune Biology, University of Debrecen, Egyetem tér, Debrecen, Hungary, 5 Institute of Medical Chemistry, Center for Pathobiochemistry and Genetics, Medical University of Vienna, Waehringerstrasse, Vienna, Austria, 6 Department of Biochemistry, Abdul Wali Khan University Mardan, Khyber Pakhtunkhwa, Pakistan, 7 Institute of Enzymology, Research Centre for Natural Sciences, Hungarian Academy of Sciences, Magyar Tudósok körútja, Budapest, Hungary, 8 Institute of Cancer Research, Medical University of Vienna, Borschkegasse, Vienna, Austria

o These authors contributed equally to this work.
* peter.chiba@meduniwien.ac.at (PC); thomas.stockner@meduniwien.ac.at (TS)

**Data Availability Statement:** All relevant data are within the manuscript and its Supporting Information files.

## Abstract

Several ABC exporters carry a degenerate nucleotide binding site (NBS) that is unable to hydrolyze ATP at a rate sufficient for sustaining transport activity. A hallmark of a degenerate NBS is the lack of the catalytic glutamate in the Walker B motif in the nucleotide binding domain (NBD). The multidrug resistance transporter ABCB1 (P-glycoprotein) has two canonical NBSs, and mutation of the catalytic glutamate E556 in NBS1 renders ABCB1 transport-incompetent. In contrast, the closely related bile salt export pump ABCB11 (BSEP), which shares 49% sequence identity with ABCB1, naturally contains a methionine in place of the catalytic glutamate. The NBD-NBD interfaces of ABCB1 and ABCB11 differ only in four residues, all within NBS1. Mutation of the catalytic glutamate in ABCB1 results in the occlusion of ATP in NBS1, leading to the arrest of the transport cycle. Here we show that despite the catalytic glutamate mutation (E556M), ABCB1 regains its ATP-dependent transport activity, when three additional diverging residues are also replaced. Molecular dynamics simulations revealed that the rescue of ATPase activity is due to the modified geometry of NBS1, resulting in a weaker interaction with ATP, which allows the quadruple mutant to evade the conformationally locked pre-hydrolytic state to proceed to ATP-driven transport. In summary, we show that ABCB1 can be transformed into an active transporter with only one functional catalytic site by preventing the formation of the ATP-locked pre-hydrolytic state in the non-canonical site.

**Funding:** We are grateful for financial support by the Austrian Science Fund (FWF; https://www.fwf.ac.at/), grant number F3509 to PC, F3524 to TS and F3525 to GS and by the Hungarian National Research, Development and Innovations Office (NKFIH; https://nkfih.gov.hu/english), grant number K124815 to KG. We are also grateful for computing time allocated to TS by the Vienna Scientific Cluster (VSC). The funders had no role in study design, data collection and analysis, decision to publish, or preparation of the manuscript.

**Competing interests:** The authors have declared that no competing interests exist.

## Author summary

ABC transporters are one of the largest membrane protein superfamilies, present in all organisms from archaea to humans. They transport a wide range of molecules including amino acids, sugars, vitamins, nucleotides, peptides, lipids, metabolites, antibiotics, and xenobiotics. ABC transporters energize substrate transport by hydrolyzing ATP in two symmetrically arranged nucleotide binding sites (NBSs). The human multidrug resistance transporter ABCB1 has two active NBSs, and it is generally believed that integrity and cooperation of both sites are needed for transport. Several human ABC transporters, such as the bile salt transporter ABCB11, have one degenerate NBS, which has significantly reduced ATPase activity. Interestingly, unilateral mutations affecting one of the two NBSs completely abolish the function of symmetrical ABC transporters. Here we engineered an ABCB1 variant with a degenerate, ABCB11-like NBS1, which can nevertheless transport substrates. Our results indicate that ABCB1 can mediate active transport with a single active site, questioning the validity of models assuming strictly alternating catalysis.

## Introduction

ABC (ATP Binding Cassette) proteins form one of the largest protein superfamilies. Most members are active transporters, which translocate their substrates across biological membranes [1, 2]. Human ABC proteins are encoded by 48 genes, which are assigned to seven subfamilies designated ABC-A to ABC-G [3, 4]. The minimal functional unit of ABC transporters comprises four domains, consisting of two nucleotide binding domains (NBDs) that hydrolyze ATP to energize transport, and two transmembrane domains (TMDs) that form the translocation pathway [2, 5, 6]. The NBDs possess a common domain architecture that is shared among all ABC proteins, while the TMDs show considerable differences [7–10]. The first high resolution structure of a full-length ABC exporter was that of the bacterial multidrug exporter Sav1866, showing a domain swapped fold with the two TMDs forming contacts with both NBS through long intracellular extensions [7]. This architecture proved prototypical for the ABCB subfamily, and was confirmed by structures of human ABCB1 [11–14], ABCB2/ABCB3 [15], ABCB8, ABCB10 [16] and ABCB11 [17]. These structures revealed three major conformations: the outward-facing conformation with associated NBDs and separated TMDs [7], the inward-facing state with separated NBDs and associated TMDs [12, 15, 16], and an intermediate conformation, in which both the TMDs and NBDs are associated [11, 14]. Binding of ATP induces NBD dimerization [18–21], while in the absence of ATP the NBDs are mobile and separated [21, 22]. Whether a functional separation is required for the transport cycle remains debated [22–26]. ATP binds to two symmetrically arranged composite nucleotide binding sites (NBSs) that are formed by both NBDs. The NBSs are formed by A-loop, H-loop, Walker A, Walker B and Q-loop of one NBD, and the X-loop and signature sequence of the other NBD [7, 27, 28]. The catalytic glutamate of Walker B and the H-loop histidine are necessary for efficient ATP hydrolysis [29–31], suggesting their direct involvement in the ATPase function. Interestingly, in about half of the human ABC transporters, the sequence of NBS1 has diverged, with "non-canonical" amino acids replacing conserved amino acids such as the catalytic glutamate in the Walker B sequence [32]. Such non-canonical NBSs are unable to hydrolyze ATP at a rate comparable to canonical NBSs, and therefore are ineffective in sustaining substrate transport [33, 34]. At present, the relevance and contribution of the asymmetric catalytic centers in substrate transport remain incompletely understood.

The human ABCB subfamily contains four full transporters, among them the multidrug resistance transporter ABCB1 [35] (MDR1 or P-glycoprotein), a major player in drug disposition, and the bile salt export pump ABCB11 (BSEP) [36]. The primary sequences of ABCB1 and ABCB11 are 49% identical, suggesting that the overall mechanics of the transport cycle may be shared. However, whereas ABCB1 has two canonical ATP binding sites, NBS1 in ABCB11 is degenerate. In ABCB1, the two NBSs are functionally equivalent and the integrity of both catalytic centers is generally believed to be needed for transport [37]. In contrast, NBS1 in ABCB11 is inactive, as it lacks the catalytic glutamate. Importantly, the degenerate NBS1 of ABCB11 contains three additional amino acids that diverge from ABCB1, but are conserved within ABCB11 sequences.

The aim of this study was to analyze the role of the non-canonical residues in the degenerate NBS1 in sustaining transport. Our results confirm that mutation of the catalytic glutamate (E556M) renders ABCB1 inactive. However, ATP hydrolysis and transport function are restored by introduction of all four diverging amino acids as present in NBS1 of ABCB11 (S474E, E556M, G1178R and Q1180E). Molecular dynamics (MD) simulations revealed that the E556M mutant is locked by tight binding of ATP, whereas transport function is restored in the quadruple mutant by weaker ATP binding and a change in the geometry of the NBD dimer. We infer from our data that the degenerate NBS sustains substrate transport by enabling ATP release without hydrolysis, thereby preventing an arrest of the catalytic cycle.

## Results

### ABCB1 and ABCB11 share an almost identical NBD interface

All sequences annotated as ABCB1 or ABCB11 homologs in the NCBI database [38] were aligned using clustalW [39] and the Gonnet PAM250 similarity matrix [40]; residue similarity was mapped onto a human ABCB1 model. As expected, the NBDs showed a higher degree of residue conservation, as compared to the TMDs. Interestingly, the NBD-NBD interface that includes the conserved NBD motifs (See S1 Fig), proved to be the region of highest sequence conservation (Fig 1), as most residues contributing to the interface are shared between ABCB1 and ACB11. The difference is limited to just four residues, located within NBS1: (i) residue E556, the catalytic glutamate of the Walker B motif of ABCB1 conforms to a methionine in ABCB11 (M584); (ii) the canonical signature sequence residues G1178 and Q1180 in ABCB1 correspond to R1221 and E1223 in ABCB11, respectively; and (iii) residue S474 in ABCB1,

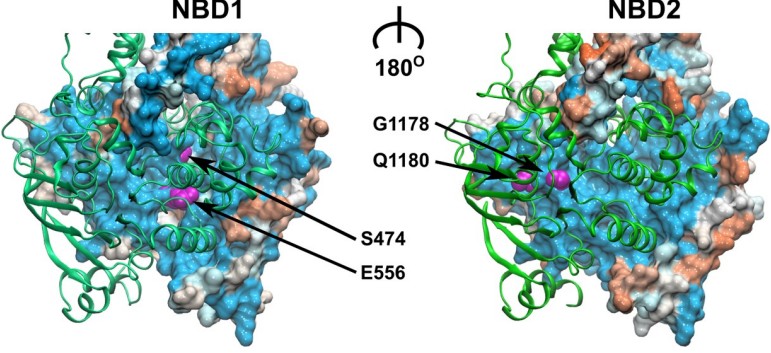

**Fig 1. Sequence conservation between ABCB1 and ABCB11 mapped onto a structural model of ABCB1.** The NBD in front is shown in green ribbon representation, while the NBD in the back shows the NBD interface in surface rendering. The surface is colored according to sequence similarity between ABCB1 and ABCB11. Blue color indicates residue identity, orange indicates maximal residue divergence. The four differing residues in NBS1 are highlighted in magenta.

which precedes the eponymous glutamine in the Q-loop, is a glutamate in ABCB11 (E502). Strikingly, the alignment of the ABCB11 sequences alone showed that these four non-canonical NBS1 residues are conserved: the residue corresponding to M584 in human ABCB11 is fully conserved; residue E1223 may also be a glutamate or a glutamine; in addition, residue I437 of the A loop is in rare cases either a serine or a threonine. This high degree of conservation indicates that evolutionary pressure remained high despite the loss of catalytic activity of the degenerate site. The residues interacting with the second NBD across the NBD-interface are shown in S2 Fig.

## ABCB1 conformation is regulated by nucleotides

To analyze the role of the four non-canonical, but conserved amino acids in the degenerate NBS1 of ABCB11, we introduced the corresponding mutations into NBS1 of ABCB1. First, we engineered a variant harboring the E556M mutation of the catalytic glutamate. To fully mimic NBS1 of ABCB11 in the context of ABCB1, a quadruple mutant was constructed that also contains the three additional diverging residues (S474E, E556M, G1178R and Q1180E). All ABCB1 variants showed comparable cell surface expression in NIH 3T3 cells, as measured by the conformation insensitive 15D3 antibody (Fig 2A) [41]. To assess the conformational flexibility of the ABCB1 variants, we used the conformation sensitive UIC2 antibody that recognizes an extracellular epitope of ABCB1 [7, 11, 13, 28, 42]. The UIC2 antibody was shown to bind with high affinity to the cyclosporin A (CsA) bound conformation, while it has low affinity for nucleotide bound conformation(s) [43, 44]. We recapitulated this change in affinity using NIH 3T3 mouse fibroblasts that express wild-type human ABCB1. UIC2 labeled approximately 30% of wild-type ABCB1 transporters, while the addition of CsA resulted in complete UIC2 labeling of cell surface expressed wild-type ABCB1. In stark contrast, the catalytic glutamate mutant was not recognized by the UIC2 antibody, despite nearly equal surface expression (Fig 2A), and the addition of CsA could only moderately promote ABCB1 recognition. Since reactivity of nucleotide-bound ABCB1 to UIC2 is generally low, these results can be explained by tight ATP binding that results in a conformational arrest in a conformation that is not recognized by UIC2 [31, 45]. Strikingly, introducing the three additional non-canonical amino acids in the quadruple mutant restored wild-type like UIC2 reactivity, suggesting that the

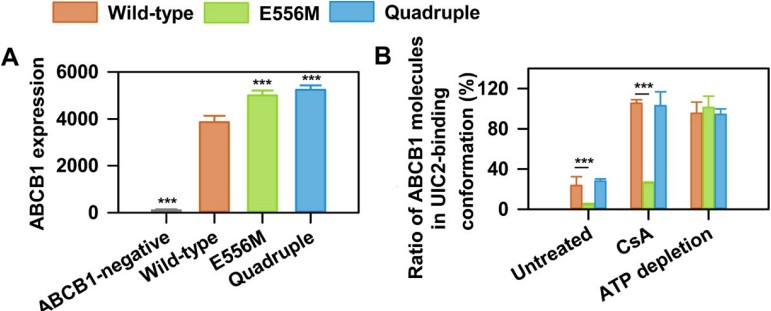

**Fig 2. UIC2 binding to ABCB1 variants carrying mutations in NBS1.** (A) Total cell surface ABCB1 expression levels were determined using the conformation insensitive 15D3 anti-ABCB1 antibody (means ± SD of n = 5 independent experiments). (B) Changes of UIC2 reactivity in wild-type ABCB1 (orange), the single (E556M, green) and the quadruple (E556M/S474E/G1178R/Q1180E, blue) mutant in response to cyclosporin A (CsA, 10 μM) treatment or ATP depletion. NIH 3T3 cells expressing the ABCB1 variants were labeled with Alexa 647 conjugated UIC2 or 15D3 mAbs (with comparable dye to antibody ratio (F/P)). The percentage of UIC2-bound ABCB1 conformers was calculated relative to the total cell surface ABCB1 expression levels determined by 15D3 labeling (mean ± SD of n = 5 independent experiments). Significant differences compared to untreated controls or wild-type are shown by *** (p<0.001).

conformational arrest can be overcome by an ABCB11-like NBS1. To show that the effect was ATP dependent, we measured UIC2 binding in ATP-depleted cells (Fig 2B) and found comparable UIC2 binding, suggesting that the studied ABCB1 variants adopt a similar conformation in the absence of ATP.

Conformational changes in the NBDs are allosterically coupled to conformational changes in the TMDs. Substrate binding to the TMDs stimulates the ATPase activity [2, 46–48], while ATP binding and hydrolysis lead to changes in the relative orientation of the TMD helices and substrate transport [44, 49, 50]. To characterize ATP-dependent conformational changes [6], we systematically varied the intracellular nucleotide concentrations in semi-permeabilized NIH 3T3 mouse fibroblasts expressing the ABCB1 variants, and determined dose-response curves of UIC2 binding in presence of increasing concentrations of MgATP or MgATP/vanadate (Fig 3). In accordance with an ATP-regulated switch in the TMD conformation, increasing MgATP concentrations decreased UIC2 staining (Fig 3A) [51, 52]. The apparent affinity of MgATP for wild-type ABCB1 ($K_A$ = 1.56 ± 0.46 mM) was found to be 3–4 fold lower than the reported $K_M$ values for ATP hydrolysis ($K_M$ = 0.3–0.5 mM) [53]. Vanadate ($V_i$), which is known to trap ABCB1 in a post-hydrolytic state by forming a ternary complex (ABCB1·ADP·$V_i$), increased the apparent nucleotide affinity of wild-type ABCB1 by 2–3 orders of magnitude ($K_A$ = 0.081 ± 0.014 mM) [47, 53–55].

Mutation of the catalytic glutamate to methionine (E556M) resulted in a markedly different behavior (Fig 3B). The apparent affinity of ATP was very high even in the absence of $V_i$ ($K_A$ = 0.077 ± 0.02 mM), while treatment with $V_i$ led to a minor increase ($K_A$ = 0.027 ± 0.001 mM), in agreement with the tight occlusion of nucleotide(s) [31, 45]. The exchange of the three additional amino acids in the quadruple mutant restored wild-type like nucleotide-sensitive conformational dynamics (Fig 3C). The dose-response curves and the apparent affinity for MgATP ($K_A$ = 1.12 ± 0.6 mM) and MgATP/$V_i$ ($K_A$ = 0.038 ± 0.001 mM) were almost indistinguishable from wild-type ABCB1 (compare Fig 3A and Fig 3C).

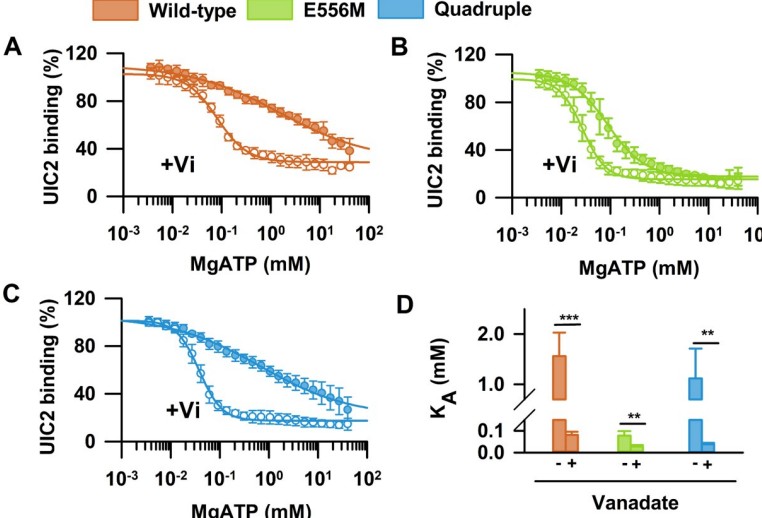

**Fig 3. MgATP dependent UIC2 binding.** NIH 3T3 cells expressing ABCB1 variants were permeabilized to allow for systematic variation of intracellular nucleotide concentrations. Dose-response curves of UIC2 binding with increasing concentrations of MgATP in the absence or presence of vanadate ($V_i$) were obtained for (A) wild-type, (B) the catalytic glutamate mutant and (C) the quadruple mutant. Dose-response curves were normalized to the UIC2 signal obtained in the presence of 0 mM ATP, which was found practically equal to the F/P corrected 15D3 signal and did not differ between the different ABCB1 variants. Panel (D) summarizes the mean apparent nucleotide affinity values ($K_A$ ± SD, n = 9–11) obtained in control and $V_i$ treated cells. Statistical comparison of the $K_A$ values of $V_i$ treated and untreated samples was carried out by unpaired Student's t-test (two-tailed test); ***p<0.001, **p<0.01.

## Mutation dependent changes of steady state ATPase activity

ABCB1 shows a basal ATPase activity that can be stimulated by substrates such as verapamil [56, 57]. The steady state ATPase activity of ABCB1 variants was quantified by measuring inorganic phosphate, a product of ATP hydrolysis, in the presence of increasing concentrations of verapamil (Fig 4). Membranes isolated from NIH 3T3 cells expressing wild-type ABCB1 showed the expected basal ATPase activity, which could be stimulated about 4-fold by the addition of verapamil (Fig 4B and 4C). In contrast, the catalytic glutamate mutant showed no ATPase activity above background levels (Fig 4B). Strikingly, the quadruple mutant showed a low steady state ATPase activity (Fig 4B) that could be stimulated by verapamil. Although the basal catalytic activity was significantly reduced, the degree of stimulation by verapamil was almost identical to wild-type, showing that drug-stimulation of the ATPase activity is restored in the quadruple mutant (Fig 4C).

## Transport activity of NBS1 mutants

The function of the multidrug exporter ABCB1 is to prevent cellular accumulation of environmental toxins through active efflux [58–60]. In addition, ABCB1 also transports many chemotherapeutic drugs and chemicals [61], including the dyes rhodamine 123 [62] and Hoechst 33342 [63]. To directly observe the effect of the mutations on transport activity, we measured cellular accumulation of Hoechst 33342 (Fig 5A) [64] and calcein (Fig 5B) in NIH 3T3 cells. Transport was quantified by the transport activity factor (TAF), corresponding to the normalized tariquidar dependent increase of Hoechst 33342 or calcein fluorescence intensities. As expected, the TAF value was close to 0 in cells devoid of ABCB1, while expression of wild-type ABCB1 effectively prevented intracellular accumulation of Hoechst 33342 (Fig 5A) and calcein (Fig 5B). Mutation of the catalytic glutamate abolished transport activity, in accordance with the loss of steady state ATPase activity and the reduced conformational flexibility. In sharp contrast, the quadruple mutant was able to limit Hoechst 33342 and calcein accumulation, though not as efficiently as wild-type ABCB1. These data clearly demonstrate that despite the absence of the catalytic glutamate, the quadruple mutant regained the ability to efflux Hoechst 33342 and calcein. We have obtained similar bell-shaped curves characterizing the

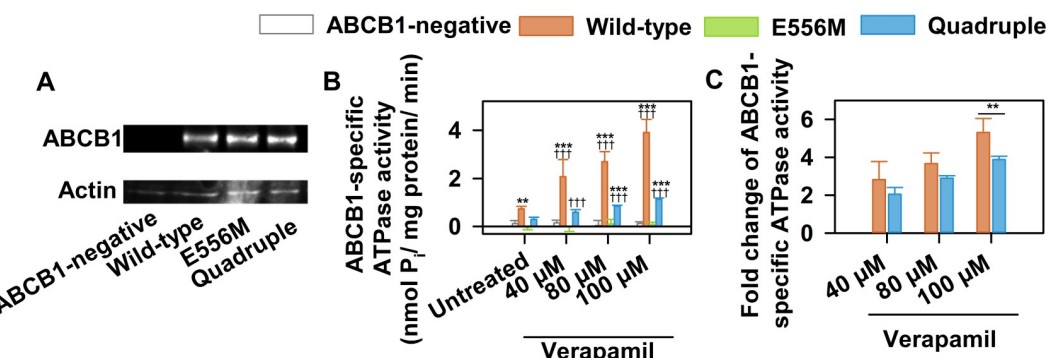

**Fig 4. Effect of the mutations on substrate-stimulated steady state ATPase activity.** (A) Expression levels of the ABCB1 variants in membranes isolated from NIH 3T3 cells, detected by Western blotting using the G-1 anti-ABCB1 mAb. (B) Steady state ATPase activity of ABCB1 variants in membranes isolated from NIH 3T3 cells. Membranes were incubated for 25 min in the presence of 3 mM MgATP and the indicated concentrations of verapamil. ABCB1 specific steady state ATPase activity is defined as valspodar inhibitable vanadate sensitive ATPase activity. Mean ± SD of 5 independent experiments are shown. (Significant differences compared to untreated NIH 3T3 control membrane samples are shown by ***: P<0.001, **: P<0.01, *: P<0.05, while significant differences between the verapamil treated and untreated samples are indicated by †††: P<0.001, ††: P<0.01, †: P<0.05). (C) Response of the ABCB1-specific steady state ATPase activities to verapamil stimulation. Significant differences between the wild-type and the quadruple mutant ABCB1 are shown by **: P<0.01.

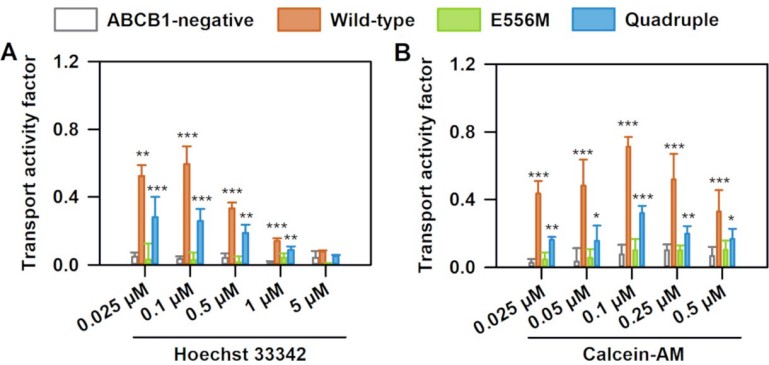

**Fig 5. Transport activity of ABCB1 variants.** ABCB1 dependent (A) Hoechst 33342 (B) and calcein-AM transport were quantified by the transport activity factor (TAF). NIH 3T3 cells expressing ABCB1 variants were incubated with Hoechst 33342 (0.025, 0.1, 0.5, 1 and 5 µM, panel A) or calcein-AM (0.025, 0.05, 0.1, 0.25, and 0.5 µM, panel B) for 30 min at 37 °C. Mean ± SD of three independent experiments are shown. Significant differences compared to ABCB1-negative cells: ***: P<0.001, **: P<0.01.

concentration dependence of transport activity in the wild-type and quadruple mutant ABCB1 variants. These bell-shaped curves were measured for many of the ABCB1 substrates, including rhodamine 123, Hoechst 33324 and calcein-AM. The ratios of the transport activity factor between wild-type and quadruple mutant are comparable for both Hoechst 33324 and calcein-AM and concentration independent, suggesting that transport specificity is unchanged for these two substrates.

## The NBS1 mutations change the molecular mechanism of ATP binding

The functional assays unequivocally showed that introduction of the degenerate NBS1 of ABCB11 in ABCB1 is compatible with drug stimulated ATPase activity and substrate transport, despite the absence of the catalytic glutamate. We performed MD simulations to elucidate the molecular mechanism underlying restored activity. Wild-type ABCB1, the catalytic glutamate and the quadruple mutant were inserted into a membrane bilayer using the membed procedure [65, 66]. Each transporter variant was independently modeled, assembled, equilibrated and simulated for 1 µs, three times. Transporters were stable in all simulations, maintaining their secondary and tertiary structures and showing similar deviations from their respective starting conformations. Detailed analysis showed that the Pγ of ATP interacts with the $Mg^{2+}$ ion, the Walker A motif and the signature sequence, while the Pα and Pβ almost exclusively interact with the Walker A motif (Fig 6) [67, 68]. Histograms of distances measured between the Cα atom of S434 (first turn of the Walker A helix) and the Pα of ATP showed equally stable interactions in the three variants, with a similar degree of fluctuations (Fig 6A). Binding of ATP to NBS1 was stable and consistent for the three replicates of wild-type ABCB1 and the E556M mutant, as measured by the root mean square deviations (RMSD) of ATP from the respective starting structure (Fig 6B). While the phosphates of ATP show equally stable binding for all three ABCB1 variants, binding of the adenosine base of ATP is less stable in the quadruple mutant. ATP shows increased dynamics in two replicates of the quadruple mutant (Fig 6B), consistent with the interpretation that binding of ATP is less strong in the quadruple mutant.

This difference in ATP dynamics correlated with the change in the overall geometry of NBS1, which deviated in the quadruple mutant. We quantified the structural changes of NBS1 and found that the distance across NBS1 between the Cα atoms of residue S434 (helix of the Walker A motif) and residue G/R1178 in the first turn of the signature sequence helix (Fig 6D)

## Stable NBD1-ATP interactions

## Changes in NBD1-NBD2 interactions

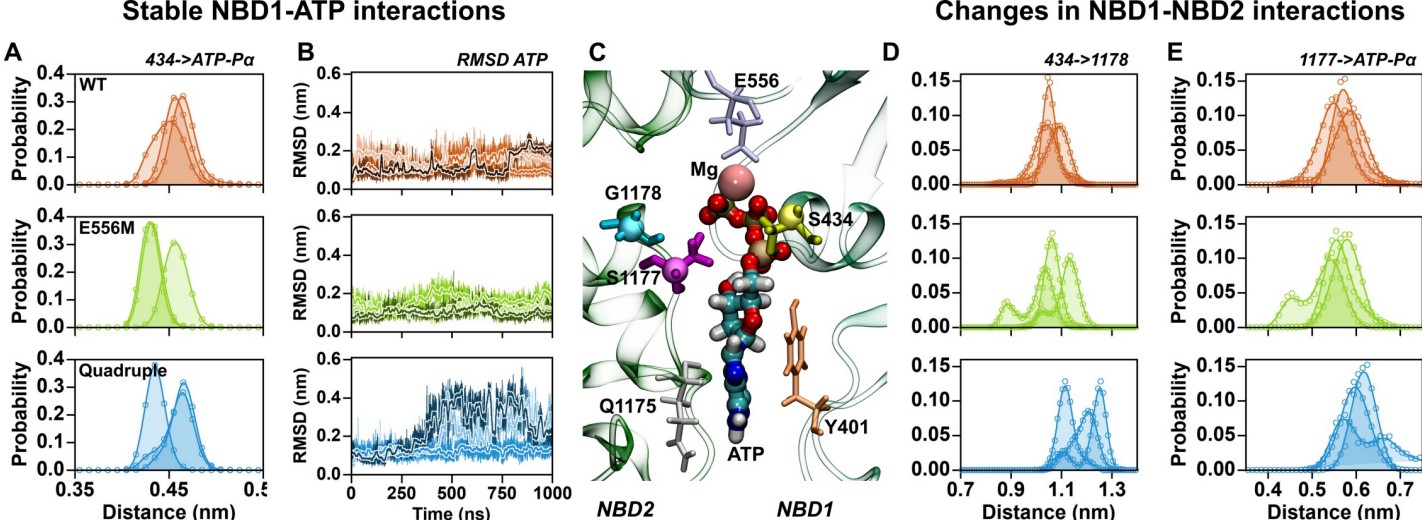

**Fig 6. ATP binding to NBS1.** (A) Distribution of distances between the Cα atom of residue S434 (Walker A) and the Pα of ATP. (B) Time evolution of the RMSD of ATP in NBS1 after fitting to NBD1. Slight variations in color are used to distinguish the three replicates. (C) Zoom into NBS1 showing the binding mode of ATP. (D) Distribution of distances between the Cα atoms of residue S434 (Walker A) and residue G/R1178 (signature sequence). (E) Distribution of distances between the Cα atoms of residue S1177 (signature sequence) and the Pα of ATP. Data from independent trajectories for wild-type simulations are shown in orange, the trajectories of the catalytic glutamate mutant are shown in green, and data for the quadruple mutant are shown in blue.

increases in the quadruple mutant. Concomitantly, the distance between the Cα atom of residue S1177 (signature sequence) and the Pα of ATP (Fig 6E) showed a slightly increased separation between ATP and the signature sequence. To bridge the wider gap, the Pγ of ATP is bent towards the more distant signature sequence, thereby maintaining simultaneous, but weaker interactions with NBD1 and NBD2.

Structural changes in NBS1 were linked to changes in structural alignment and dynamics of the ATP-bound NBD dimer. We used the distance between the signature sequences of NBD1 and NBD2 and the distance between the Walker A motif and the signature sequence across NBS1 as a measure to quantify global structural changes linked to NBD1-NBD2 association (Fig 7A). Wild-type ABCB1 showed the largest degree of variation in both distances and therefore the highest dynamic range. The catalytic glutamate mutant showed a smaller dynamic range and populated only shorter distances, consistent with a single conformation of a conformationally locked state. Similarly, the quadruple mutant sampled a smaller range of shorter distances. Fig 7A shows a simultaneous reduction in the dynamic range of NBD dimer geometry and NBS1 closure, indicating that mutation-associated local differences in NBS1 are correlated with the global conformational changes in the NBD dimer.

The Q-loops connect the NBSs with intracellular loop 2 and 4 of the TMDs and therefore play a role in the cross-talk between ATP hydrolysis and the TMDs. Single Q-loop mutants (Q475A or Q1118A) were shown to maintain almost wild-type like transport function and UIC2 binding. In contrast, the double mutant (Q475A/Q1118A) lost ATPase activity and transport capability, and the TMDs were no longer able to adopt a nucleotide bound conformation reminiscent of wild-type ABCB1 [69]. We observed a strong correlation between the separation of the two signature sequences and the distance between the signature sequence of NBD2 and the Q-loop of NBD1 (Fig 7B) for wild-type ABCB1. The E556M mutant samples mainly conformations that overlay with the shortest conformations detected for wild-type ABCB1. Also conformations at larger distances are sampled, but their probability is much lower. Thus, the correlation seems much weaker in the catalytic glutamate mutant, presumably because of the close proximity of the E556M mutation to the Q-loop and due to the restrained

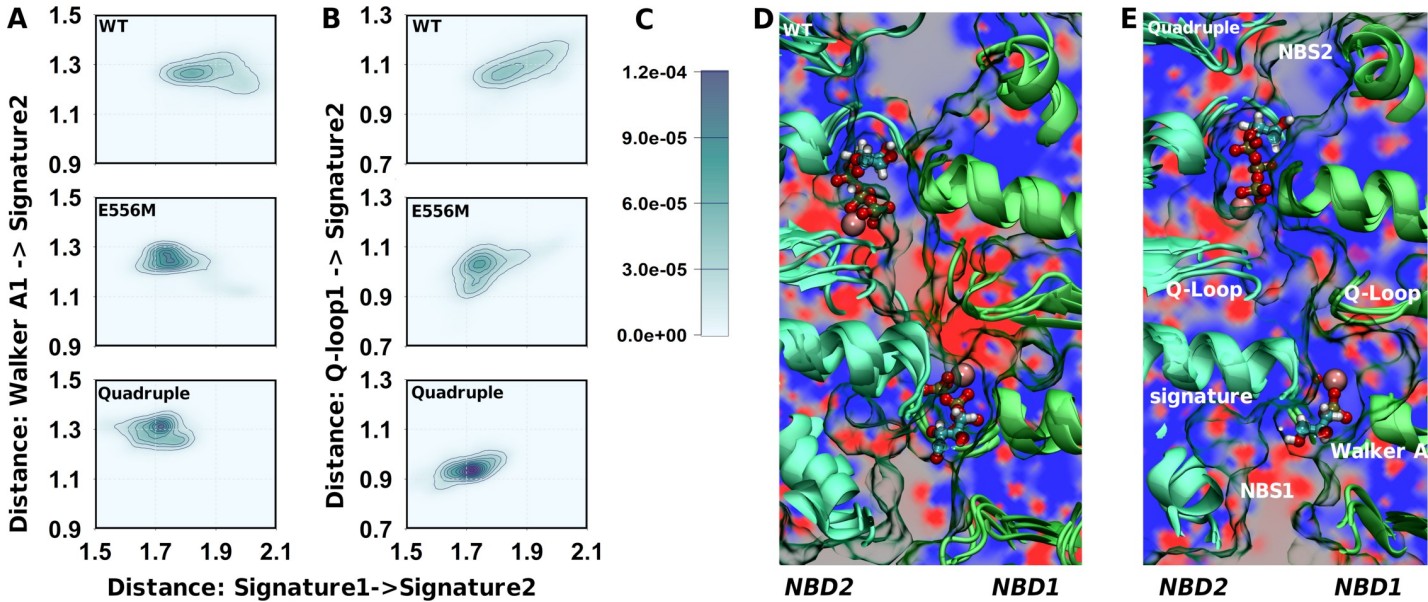

**Fig 7. Structural changes in the NBD interface associated with NBS1 mutations.** (A) Two dimensional probability distribution of distances: the x-axis reports on the distance between the signature sequences of NBD1 and NBD2, the y-axis shows the distance across NBS1 between the Walker A motif (residue 434) and the signature sequence in NBS1 (residue 1178). (B) Two dimensional probability distribution of the distance within NBS1: the x-axis reports on distances between the signature sequences of NBD1 and NBD2, the y-axis measures distance within NBS1 between the Q-loop of NBD1 (residue 475 and 476) and the signature sequence in NBS1 (residue 1178). (C) Probability scale for panel A and B. (D) Visualization of the electrostatic potential (calculated using APBS without the presence of ATP and $Mg^{2+}$) in a plane perpendicular to the main axis of ABCB1 and intersecting with both signature motifs. The electrostatic potential was averaged over the replicates of wild-type ABCB1 simulations (E) Electrostatic potential of the quadruple mutant as determined in panel C.

conformational dynamics. The quadruple mutant was characterized by a collapsed dynamic range in the signature to Q-loop distance and an almost complete loss of the above correlation, indicating a change in structure and dynamics of the NBD dimer. Importantly, the mean distance between the Q-loop and the signature sequence is shorter, consistent with a change in the local geometry of NBS1.

The studied ABCB1 variants include mutations that change amino acid charges. A methionine neutralizes the negative charge of the catalytic glutamate (E556M), while the quadruple mutant adds a positive charge (G1178R) and two negative charges (S474E and Q1180E). An important cumulative effect is a change in the sign of the overall electrostatic potential between the Q-loops in the center of the NBD dimer. The key difference is the G1178R mutation, which places its positively charged guanidinium group between the Q-loops. The change in the electrostatic potential extended to the signature sequence and to the Pγ of ATP in NBS1 (Fig 7C and 7D), thereby also affecting electrostatic interactions between the NBDs and MgATP. Importantly, in addition to its electrostatic effects, the guanidinium group of G1178R also assumes a strong structural role by forming hydrogen bonds with the backbone carbonyl oxygens of the two Q-loop glutamines Q475 and Q1118 (Fig 8A and 8D) thereby acting as a spacer that restrains the distances between the Q-loops (Fig 8A). The main structural consequence is a more defined structural arrangement of the NBDs.

Crystal structures revealed that in the ATP bound state, the D-loops are in close proximity to each other and to the Pγ of ATP. Biochemical data showed that transport function is strongly modulated by D-loop mutations in ABCB11 [70]. Also, a D674A mutation converted the transporter associated with antigen processing (TAP) from a strictly unidirectional transporter to a nucleotide-gated facilitator [71]. In TM287/288, mutation of the D-loop aspartate in the consensus site strongly reduced ATPase activity, while having little effect on the ATPase

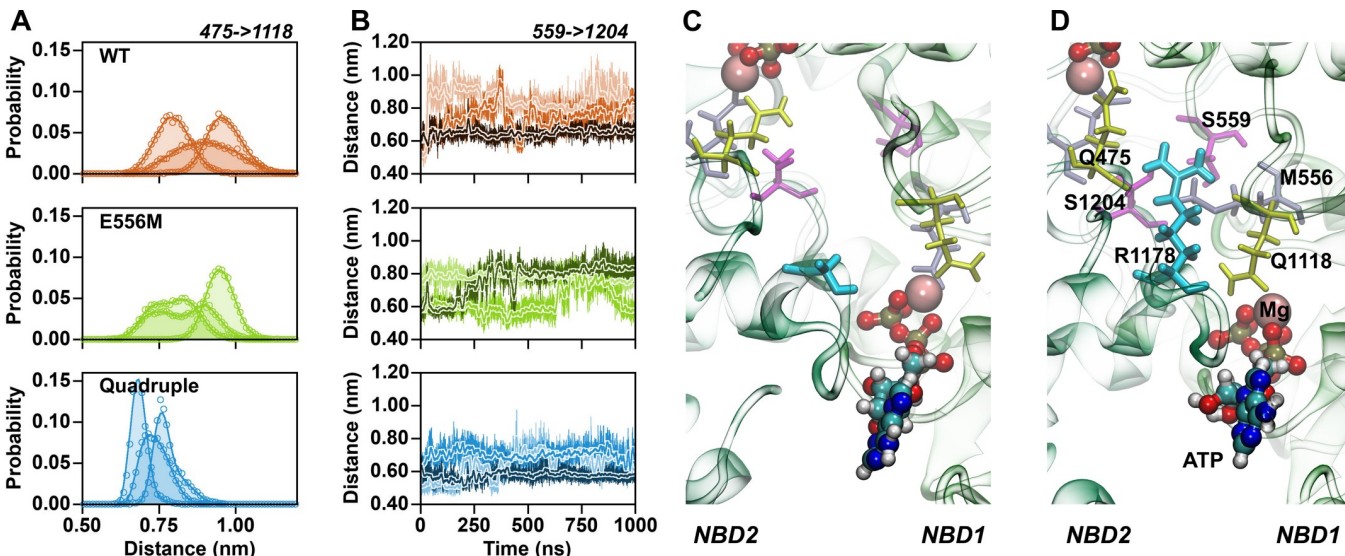

**Fig 8. The guanidinium moiety of G1178R interacts with both Q-loops.** (A) Distribution of distances between the Q-loop carbonyl oxygens of Q475 and Q1118. Data from independent trajectories for wild type simulations are shown in orange, the trajectories of the catalytic glutamate mutant are shown in green, and data for the quadruple mutant are shown in blue. (B) Time evolution of distances between the Cα atoms of the D-loop serines S559 and S1204. (C) Relative D-loop orientation shown for wild-type ABCB1, highlighted by D-loop serines S559 and S1204 (magenta). (D) Representative position of the guanidinium group of G1178R (cyan) forming hydrogen bonds with the backbone carbonyl of Q-loop glutamines (yellow).

activity in the degenerate site [72]. The time evolution of the distance between the D-loop Cα atoms of S559 and S1204 (Fig 8B) showed that their relative distance was dynamic for wild-type and the catalytic glutamate mutant. The comparison of the simulations of the three replicas showed that different interactions were formed, but these interactions were not very stable, because changing within the time-window of the 1 μs long trajectories. In contrast, the D-loop distance was more restrained in the quadruple mutant. The dynamics (Fig 8B) were limited and sampled shorter distances, while side chains of the two D-loops were always in direct contact.

## Discussion

Substrate transport by ABCB1 is the integral result of a sequence of molecular events that include ATP binding and hydrolysis, drug binding and drug release. ATP hydrolysis by the NBDs converts the chemical energy stored in the phosphate bonds of ATP into NBD motions that are propagated to the TMDs through conformational cross-talk. Evidence for the inter-domain cross-talk comes from i) crystal structures showing correlated conformational changes in TMDs and NBDs [7, 42]; ii) substrate-stimulated ATPase activity, revealing that substrate-induced conformational changes in the TMDs are propagated to the NBDs [46–48]; iii) biochemical experiments showing that substrate binding to the TMDs leads to conformational changes in the NBDs [23, 26, 49, 73]. Our data reveal that nucleotides are the main determinants of the overall ABCB1 conformation, as nucleotide binding (ATP, ADP or ADP + $V_i$) shapes transporter conformation in a mutation dependent manner, while ABCB1 variants were equally well recognized by the UIC2 antibody in the absence of nucleotides. Accordingly, ABCB1 remains locked in a pre-hydrolytic ATP-bound state, when hydrolysis is prevented or slowed down to an undetectable level by the E556M mutation. ATP is the primary nucleotide controlling transporter conformation in energized cells, because within cells its concentration is typically 10 fold above the $K_D$ for ABCB1 [74] and an order of magnitude higher than the

ADP concentration. In agreement with our data, LRET [23], FRET [75] and EPR [73] experiments showed that the NBDs assume larger inter-domain distances in the absence of ATP.

The mechanism of ATP hydrolysis remains debated and several catalysis models have been proposed [21]. However, all currently proposed models suggest a prominent role for the catalytic glutamates (E556 and E1201 in ABCB1), in line with the deleterious effect of mutations affecting this residue in basically all investigated ABC transporters. The exact role of the catalytic glutamate in ATP hydrolysis remains disputed. Higher resolution structures of ABC transporters have been obtained in the presence of ATP only when the catalytic glutamate E556 is mutated, supporting the notion that in absence of the glutamate transporters adopt a conformationally restrained state allowing ATP hydrolysis to proceed only at a low rate. A direct contact between the negatively charged catalytic glutamate E556 and the negatively charged phosphate moieties of ATP is not postulated in functional models. A nucleophilic attack on the phosphate atoms leading to bond breakage between the β and γ-phosphate of ATP may be mediated by an activated OH⁻ nucleophile generated with the contribution from residue E556 [76, 77]. It has been shown that the E556Q mutant has very low, but detectable residual ATPase activity in detergent purified ABCB1 [31]. This low ATPase activity is not sufficient to sustain any detectable net transport. In our hands, the steady-state ATPase activity of the E556M mutant does not show any statistically significant difference from controls (Fig 4B). It is generally believed that inactivation of one NBS blocks the transport function of symmetric ABC transporters [31, 33]. Yet a large number of human ABC transporters, including ABCB11, harbor a degenerate NBS1 [32, 70]. These transporters are active, despite missing a catalytic glutamate. Maintained evolutionary pressure on NBS1 of ABCB11 suggests that the non-canonical NBS has acquired a novel function that allows for overcoming the conformational arrest without the need for ATP hydrolysis [70]. To investigate the relevance of conserved amino acids aligning the degenerate NBS1, we created the transport deficient catalytic glutamate mutant (E556M) in ABCB1 and also inserted the complete set of non-canonical amino acids of ABCB11 into ABCB1 (S474E, E556M, G1178R and Q1180E). We find that ABCB1 becomes transport incompetent, conformationally trapped and ATP hydrolysis deficient, when only the catalytic glutamate is mutated, confirming earlier findings [31, 70]. Strikingly, inserting the complete non-canonical NBS1 of ABCB11 into ABCB1 restored conformational dynamics, substrate stimulated ATPase activity and transport function, most likely by preventing ATP occlusion in NBS1 and energizing transport by hydrolysis only in NBS2. These results also challenge models assuming strictly alternating catalysis [37, 78], proposing the continuous switching of ATP hydrolysis between NBS1 and NBS2 of ABCB1 [79]. While not very likely, it remains a possibility that wild-type ABCB1 hydrolyses two or more ATP molecules per transport cycle in one NBS to sustain a higher rate of substrate transport compared to the quadruple mutant. Experimental data [22, 45] showed that ABCB1 occludes only one ATP in an asymmetric state, suggesting that only one ATP is primed for hydrolysis at a time. Our results argue against a model which predicts that a second ATP hydrolysis event in the second NBS is needed to reset the transporter, because in the quadruple mutant only NBS2 is catalytically active at a high turnover.

We used MD simulations to probe into the molecular origin of the restored transport function. We previously showed for wild-type ABCB1 that ATP binding stabilizes the NBD dimer by ~42 kJ/mol and that the hydrolysis products ADP and inorganic phosphate create a high energy state that induces NBD separation [67, 68]. Our simulations revealed that binding of ATP and its interactions with NBS1 are similar for wild-type ABCB1 and the catalytic glutamate mutant, but showed more restricted dynamics in the E556M mutant, indicating stronger interactions that are consistent with a locked state. ATP cannot be efficiently released, because the free energy of ATP binding shows a deep energy well that stabilizes the ATP-bound conformation [67, 68]. Therefore, ATP remains tightly bound in the E556M ABCB1 variant, as

efficient hydrolysis is prevented, thereby leading to an arrest of the transport cycle. Simulations show that creating an ABCB11-like NBS1 in ABCB1 prevents this occlusion, thereby avoiding the locked state observed for the E556M mutant. The quadruple mutant includes the same E556M residue change, but these additional residue changes lead to a change in the geometry of NBS1 (Fig 7), effectively weakening ATP binding and preventing occlusion (Fig 6). We can infer from these data that the ABCB11-like NBS1 of the quadruple mutant allows for ATP binding, but it prevents occlusion through altered interactions, thereby avoiding strong binding of ATP. Our results support random recruitment of the two catalytic centers for ATP hydrolysis in ABCB1 [51]. The quadruple mutant is reminiscent of the degenerate NBS1 of ABC transporters. Future work will reveal if in the quadruple mutant ATP remains loosely bound to the degenerate NBS1 throughout the transport cycle, keeping the two NBDs in close proximity, or alternatively, if ATP unbinds without hydrolysis from NBS1 in every transport cycle, as suggested for the yeast ABC transporter Pdr5 [80, 81].

## Materials and Methods

### Sequence alignments and homology modeling

Sequences of the human ABCB1 and ABCB11 were aligned using clustalW (v.2.0) [39]. The homology model of ABCB11 was created using Sav1866 as a template using MODELLER software (version 9v12) [82]. Model quality was evaluated using Ramachandran plots and by applying the DOPE score of energetic evaluation [83]. The best model of ABCB11 was used for analysis. The data driven model of ABCB1, based on the Sav1866 crystal structure, has been described previously [84].

### Simulation

Production runs were carried out using the Gromacs simulations package, version 5.1.4. Berger lipids [85] were used for describing the POPC membrane. The AMBER force field [86] was used for the protein, the parameters for ATP have been described in [68, 87]. Temperature was maintained at 310 K using the v-rescale ($\tau = 0.1$ ps) thermostat [88], while separately coupling protein, membrane and solvent. Pressure was maintained at 1 bar using the Berendsen barostat [89]. The pressure coupling constant was set to 1.0 ps, the compressibility to $4.5 \times 10^{-5}$ bar$^{-1}$. Long range electrostatic interactions were described using the smooth particle mesh Ewald method [90] applying a cutoff of 1.0 nm. The van der Waals interactions were described using the Lennard Jones potential applying a cutoff of 1.0 nm. Long range corrections for energy and pressure were applied. The bonds and angles of the water molecules were constrained using the SETTLE algorithm [91], while all other bonds were constrained by LINCS [92].

### Antibody purification and labeling

The UIC2 and 15D3 anti-ABCB1 mAbs were prepared from hybridoma supernatants using affinity chromatography and were >97% pure by SDS/PAGE [93]. Hybridoma cell lines were obtained from the American Type Tissue Culture Collections (Manassas, VA, USA). The UIC2 and 15D3 antibodies were labeled with Alexa 647 succinimidyl ester (A647; Life Technologies, Inc., Carlsbad, CA, USA) and separated from the unconjugated dye by gel filtration on a Sephadex G-50 column.

### Cell lines

The NIH 3T3 mouse fibroblast cell line was a kind gift from Michael Gottesman (National Institutes of Health, Bethesda, MD). The cells were grown as monolayer cultures at 37˚C in an

incubator containing 5% $CO_2$, and were maintained by regular passages in Dulbecco's modified Eagle's medium (DMEM, Sigma-Aldrich, Budapest) supplemented with 10% heat-inactivated fetal calf serum, 2 mM L-glutamine, and 0.1 mg/ml penicillin-streptomycin cocktail.

## Vector constructs

Sleeping Beauty transposon vectors containing the wild-type, E556M, and the quadruple mutant (S474E, E556M, G1178R, Q1180E) human MDR1 cDNA were constructed. Site-directed mutagenesis was performed using the QuikChange II Site-Directed Mutagenesis Kit (Agilent Technologies, Santa Clara, CA, USA) on the pAcUW-LMDR1 vector carrying the wild-type human ABCB1 cDNA. Mutations were generated according to the manufacturer's instructions. Full-length ABCB1 cDNAs were sequenced and mutations were confirmed in all SB constructs.

## Establishment of transgenic cell lines

The NIH 3T3 cell clones stably expressing the wild-type and mutant human ABCB1 transporter variants were established by the Sleeping Beauty transposon-based gene delivery system [94, 95], using the 100 fold hyperactive SB transposase [96, 97]. NIH 3T3 mouse fibroblasts were co-transfected with the SB transposase and SB transposon vector constructs by Lipofectamine 2000 reagent (Life Technologies, Budapest, Hungary), in accordance with the manufacturer's instructions. Briefly, $3 \times 10^5$ cells were seeded in 6-well-plates, 24 hours later cells were transfected with 2 μg vector DNA per well in a 10:1 ratio for the SB transposon and transposase constructs. 48 hours after transfection transgene positive cells were sorted by flow cytometry (Becton Dickinson FACSAria III Cell Sorter (Becton Dickinson, Mountain View, CA, USA)) based on the cell surface expression of ABCB1. Protein expression was measured by antibody labeling using the human ABCB1-specific monoclonal antibodies MRK16 (Abnova GmbH Heidelberg, Germany) or 15D3. To obtain homogeneously expressing cell populations, the sorting procedure was repeated 2 or 3 times.

## Permeabilization of cells with streptolysin O

NIH 3T3 cells ($1 \times 10^7$ cells/ml) were treated with 300 U/ml streptolysin O (SLO, Sigma-Aldrich, Budapest) in the presence of 10 mM DL-Dithiothreitol (DTT), protease inhibitor cocktail (Sigma-Aldrich, Budapest, Hungary) and 100 μg/ml phenylmethanesulfonyl fluoride (PMSF) in phosphate-buffered saline (PBS) containing 1% fetal bovine serum (FBS-PBS) at 37˚C for 30 min, allowing permeabilization of approximately 60–80% of the cells (judged by propidium iodide (PI) staining). The reaction was stopped with 10 ml ice-cold FBS-PBS and the cells were centrifuged for 5 min at $525 \times g$ at 4˚C. Unbound toxin was removed by washing the cells 3 times with FBS-PBS and the cell pellet was re-suspended in FBS-PBS.

## Determination of the apparent affinity of nucleotide binding

Apparent affinity of nucleotide binding ($K_A$) was determined as it is described in Barsony et al. [51]. Briefly, MgATP was added in a broad concentration range to permeabilized cells ($1 \times 10^6$ cells/ml) in the presence or absence of 0.5 mM vanadate ($V_i$) for 30 min at 37˚C. Then the samples (without washing) were further incubated with 10 μg/ml UIC2-A647 mAb for another 30 min at 37˚C. Following antibody labeling, samples were washed 3 times with ice-cold FBS-PBS and re-suspended in ice-cold FBS-PBS containing 3 μg/ml PI. The UIC2-A647 fluorescence intensity of the PI positive cells was measured by flow cytometry and plotted as a function of the nucleotide concentration. To determine the apparent affinity of ABCB1 to the

nucleotides ($K_A$) data points were fitted with the four-parameter Hill function, where $F_{min}$ and $F_{max}$ values are the minimum and maximum fluorescence intensities.

## UIC2 reactivity assay

Cells ($5 \times 10^5$ cells/ml) were pre-treated with 10 μM CsA in PBS supplemented with 8 mM glucose (gl-PBS) for 30 min at 37°C. ATP depletion was induced by Na-azide (10 mM) and 2-deoxy-D-glucose (8 mM) treatment for 30 min at 37°C in glucose-free PBS. The ABCB1 reactive mAbs UIC2-A647 (10 μg/ml) or 15D3 (30 μg/ml) were added directly without washing step and cells were further incubated for 30 min at 37°C. Then the samples were washed twice with ice cold PBS and re-suspended in 250 μl ice cold PBS before flow cytometric analysis. The A647-conjugated UIC2 and 15D3 antibody stocks had comparable dye to antibody ratios (F/P = 1.75–2.5) and the fluorescence intensities were corrected for F/P values. UIC2-reactivity (i.e. the percentage of cell surface ABCB1 molecules in a UIC2 reactive conformation) was calculated as a ratio of the F/P-corrected UIC2 and 15D3 signals.

## Preparation of membranes

Membrane fraction of NIH 3T3 mouse fibroblast cells expressing wild-type, E556M and quadruple mutant ABCB1 and their ABCB1-negative counterpart was prepared according to Sarkadi et al. [56] with minor modifications previously described in [98]. Cells were harvested by scraping them into ice-cold PBS and washed twice at 300 × g for 5 min. Subsequently, cells were lysed and homogenized in TMEP solution (50 mM Tris, pH = 7.0, with HCl), supplemented with 50 mM mannitol, 2 mM EGTA, 0.5 mM phenylmethylsulphonyl fluoride (PMSF) and protease inhibitor cocktail (Sigma-Aldrich, Budapest)) at 4°C using a glass tissue homogenizer. Nuclear debris and intact cells were selectively removed by centrifugation at 500 × g for 10 min at 4°C. Supernatant was further centrifuged for 60 min at 28,000 × g at 4°C and the membrane pellet was re-suspended in TMEP solution. Membrane samples were stored at -80°C. Protein concentration of the membrane samples was determined by the Lowry method [99].

## Western blot analysis

Cell membrane samples (5 μg/slot) were subjected to SDS-polyacrylamide gel electrophoresis on 10% polyacrylamide gel and were electroblotted to 0.45 μm pore size nitrocellulose membrane (GE Healthcare Life Sciences, Little Chalfont, Buckinghamshire, UK). ABCB1 expression was detected by a monoclonal anti-ABCB1 antibody (G-1, Santa Cruz Biotechnology Inc., Santa Cruz, CA, USA), while as a loading control actin expression was detected by a monoclonal anti-actin antibody (C-2, Santa Cruz Biotechnology Inc., Santa Cruz, CA, USA) at 1:5,000 dilution. As a secondary antibody a goat anti-mouse HRP-conjugated IgG (Santa Cruz Biotechnology Inc., Santa Cruz, CA, USA) was applied at 1:5,000 dilution. Images were collected using a FluorChem Q Alpha Innotech imaging system.

## ATPase activity measurements

The ABCB1-specific ATPase activity of the membrane samples was determined by measuring the amount of inorganic phosphate ($P_i$) released in the ATPase reaction with modifications described in [51, 56, 100]. Membrane samples (15 μg membrane protein/sample) were pre-incubated in 60 μl ATPase assay premix (50 mM MOPS, 65 mM KCl, 6.5 mM $NaN_3$, 2.6 mM DTT, 1.28 mM ouabain, 0.65 mM EGTA, pH = 7.0) in the presence or absence of 100 μM $Na_3VO_4$ (vanadate) and 5 μM of the ABCB1 inhibitor valspodar (Sigma-Aldrich, Budapest) at

37°C. The ATPase reaction was started with the addition of 3.2 mM MgATP. After 25 min incubation at 37°C, the reaction was stopped by 40 μl 5% SDS, then the samples were incubated with 105 μl color reagent [101] at room temperature for 30 min. Absorbances of the samples were measured at 700 nm using a BioTek Synergy HT plate reader (BioTek Instruments, Winooski, VT, USA) and the amount of released $P_i$ was calculated. To increase the signal to noise ratio of the ATPase assay, values were corrected for the background activity associated with other endogenous ATPases. Thus, ABCB1-specific ATPase activity is defined as the difference between the vanadate-sensitive ATPase activities measured in the absence and the presence of valspodar.

## Hoechst 33342 and calcein accumulation studies

For measurement of the transport activity of the mutant ABCB1 variants a cell based Hoechst 33342 and calcein-AM accumulation assay was used [64, 102]. Cells ($5 \times 10^5$ ml$^{-1}$) in gl-HEPES (20 mM HEPES, 123 mM NaCl, 5 mM KCl, 1.5 mM MgCl$_2$, 1 mM CaCl$_2$, 7 mM glucose) for Hoechst 33342 assay or in gl-PBS for calcein-AM assay were pre-incubated in the presence or absence of specific ABCB1 inhibitor (tariquidar (1 μM)) for 10 min at 37°C and then were loaded with different concentrations of Hoechst 33342 (5.0, 1.0, 0.5, 0.1 and 0.025 μM) and calcein-AM (0.5, 0.25, 0.1, 0.05 and 0.025 μM) for 30 min. The samples were washed with ice-cold HEPES or PBS buffer containing 0.5% FBS at $300 \times g$ for 5 min and were kept on ice until flow cytometric measurement. Dead cells were excluded from the analysis on the basis of PI staining. The transport activity of the ABCB1 variants was described by the Transport Activity Factor (TAF) calculated according to the following formula: TAF = $(MF_{inh}-MFI_0)/MFI_{inh}$, wherein $MFI_{inh}$ and $MFI_0$ are the mean fluorescence intensity values measured in the presence and absence of ABCB1 inhibitor, respectively [102].

## Flow cytometry

Intracellular Hoechst 33342 and calcein accumulation was measured using a Becton Dickinson FACSAria III Cell Sorter (Becton Dickinson, Mountain View, CA, USA). Hoechst 33342 was excited with a 365 nm UV laser and the emitted blue light was detected using a 445/40 band pass filter. Calcein was excited with a 488 nm blue laser and fluorescence was detected using a 502 long pass dichroic mirror and a 530/20 band pass filter. PI was excited by the 562 nm line of a solid-state laser and the emitted light was detected applying a 590 nm dichroic mirror and a 595/50 nm band-pass filter.

UIC2-A647 and 15D3-A647 labeling of cells was measured by using a Becton Dickinson FACS Array (Becton Dickinson, Mountain View, CA, USA) flow cytometer. A 635 nm laser was used for the excitation of the Alexa 647 dye and the fluorescence was detected in the red channel (661/16 nm), while the 532 nm laser was used for the excitation of PI (detected at 585/42 nm). Cell debris was excluded from analysis on the basis of FSC and SSC signals. Cytofluorimetric data were analyzed by using FCS Express 4 Research Edition (De Novo Software, Glendale, CA, USA).

## Statistical analysis

Data were analyzed using SigmaStat (version 3.1, SPSS Inc., Chicago, IL, USA) and are presented as means ± SD. Comparison of two groups was carried out by unpaired *t*-test, statistical significance in the case of three or more groups was assessed using analysis of variance (ANOVA), applying the Holm-Sidak multiple comparison test for *post-hoc* pair-wise comparison of the data. In the case of unequal variances Dunnett T3 *post-hoc* pair-wise comparison method was used. Differences were considered significant at $P<0.05$.

## Supporting information

**S1 Fig. Visualization of NBD motifs: The NBD-NBD interface of a single NBD is shown in cartoon representation.** The known motifs are highlighted by color, ATP is show in yellow, the $Mg^{2+}$ ions in pink. The arrows in the lower left corner indicates the orientation of ABCB1 relative to the membrane. The green and red arrows are oriented parallel to the membrane plane, the blue arrow indicates the direction perpendicular to the membrane.
(TIF)

**S2 Fig. Residues contact across the NBD dimer interface of wild-type ABCB1. (A)** Cartoon representation of NBD1 of ABCB1, oriented as in supporting S1 Fig. ATP is shown as yellow sticks, $Mg^{2+}$ as a pink sphere. Residues in direct contact with NBD2 are indicated by light green sphere of their respective Cα atoms. Residue 556 is highlighted in red. (**B**) Cartoon representation of NBD2 of ABCB1, shown similar as NBD1 in panel A. Residues directly interacting with NBD1 are highlighted by dark green sphere of their respective Cα atoms. Residues 1178 and 1180 are indicated in red. (**C**) Top view of the NBD dimer showing the Cα atoms of all residues that are in direct contact. (**D**) The matrix summarize all contacts across the NBD-dimer interface, indicating in red the residues mutated in the quadruple mutant. ATP acts as glue [21, 67] and interaction hub, while shielding residues from direct interaction.
(TIF)

## Author Contributions

**Conceptualization:** Katalin Goda, Gergely Szakács, Peter Chiba, Thomas Stockner.

**Data curation:** Katalin Goda, Yaprak Dönmez-Cakil, Szabolcs Tarapcsák, Thomas Stockner.

**Formal analysis:** Yaprak Dönmez-Cakil, Szabolcs Tarapcsák, Gábor Szalóki, Dániel Szöllősi, Zahida Parveen, Dóra Türk.

**Funding acquisition:** Katalin Goda, Gergely Szakács, Peter Chiba, Thomas Stockner.

**Investigation:** Yaprak Dönmez-Cakil, Szabolcs Tarapcsák, Gábor Szalóki, Dániel Szöllősi, Zahida Parveen, Dóra Türk.

**Methodology:** Katalin Goda, Yaprak Dönmez-Cakil, Gergely Szakács, Peter Chiba, Thomas Stockner.

**Project administration:** Katalin Goda, Thomas Stockner.

**Resources:** Katalin Goda, Gergely Szakács, Peter Chiba, Thomas Stockner.

**Supervision:** Katalin Goda, Gergely Szakács, Peter Chiba, Thomas Stockner.

**Validation:** Katalin Goda, Szabolcs Tarapcsák, Thomas Stockner.

**Visualization:** Szabolcs Tarapcsák, Dániel Szöllősi, Thomas Stockner.

**Writing – original draft:** Katalin Goda, Yaprak Dönmez-Cakil, Thomas Stockner.

**Writing – review & editing:** Katalin Goda, Gergely Szakács, Peter Chiba, Thomas Stockner.

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
