## [Decision Letter · Decision Letter 0]

13 Nov 2019

Dear Dr Stockner,

Thank you very much for submitting your Research Article entitled 'Human ABCB1 with an ABCB11-like degenerate nucleotide binding site maintains transport activity by avoiding nucleotide occlusion' to PLOS Genetics. Your manuscript was fully evaluated at the editorial level and by independent peer reviewers. The reviewers appreciated the attention to an important problem, but raised some substantial concerns about the current manuscript. Based on the reviews, we will not be able to accept this version of the manuscript, but we would be willing to review a revised version.

Should you decide to revise the manuscript for further consideration here, your revisions should address the specific points made by each reviewer. We will also require a detailed list of your responses to the review comments and a description of the changes you have made in the manuscript. We apologize at this point for the delayed feedback on the article but as you will see from the reviewer comments the detailed assesment required time.

If you decide to revise the manuscript for further consideration at PLOS Genetics, please aim to resubmit within the next 60 days, unless it will take extra time to address the concerns of the reviewers, in which case we would appreciate an expected resubmission date by email to plosgenetics@plos.org.

[LINK]

Yours sincerely,

Thorben Cordes

Guest Editor

PLOS Genetics

Gregory Barsh

Editor-in-Chief

PLOS Genetics

Reviewer's Responses to Questions

**Comments to the Authors:**

Reviewer #1: The manuscript by Goda, Donmez-Cakil, Chiba, Stockner and coworkers attempts to address a critical biological question regarding the nucleotide-dependent conformational changes needed to occur within the Nucleotide-binding domains (NBDs) and subsequently to be transmitted to the translocator (TMDs) for transport to be achieved. NBDs represent the most conserved domains of the largest group of transport proteins known (ABC proteins) that are found in all kingdoms of life, and are also present in transport unrelated ABC proteins. Numerous models have been proposed to address the means by which the nucleotide occupancy state (ATP, ADP, etc) and hydrolysis affects the conformational states of the NBDs and the cross-talk of the NBD-TMD with changes occurring at the TMDs (e.g. substrate binding to TMDs). It is crucial to address if there are common -universal- principles governing the function of all ABC proteins, or completely different models describe the function of different ABC proteins. To answer such questions the investigators compared 2 “flavors” of NBDs found in ABC exporters: Those that form 2 canonical Nucleotide binding sites (NBS) and those having one degenerate NBS. Wisely, to accomplish this, they selected 2 highly related proteins (ABCB1 and ABCB11, sequence identity:49%), as in such a case it is possible to address if it is achievable to “transform” one category to the other (by pinpointing the differences), what is needed for this conversion and the concomitant structural and functional repercussions. The NBS1 of ABCB1 has been transformed to the degenerate NBS1 of ABCB11 by introducing a mutation in the conserved E of Walker B motif and explored the structural and functional repercussions of this transformation. This transformation was proved to be deleterious, but the authors managed to create a gain of function derivative in which the NBD-NBD interface was “copied” to resemble that of ABCB11.

A combination of biochemical/functional experiments with dynamic simulations yielded important mechanistic insights that should be published. However, to evaluate publication in PLOS Genetics, a part of the conclusions needs to be reconsidered, as contradictory to published experimental data. Moreover, additional analysis is needed to reach the conclusions that would allow publication in PLOS Genetics. Finally, some figures and text should be adapted to render the conclusions clear and straightforward to the readers.

1. Fig.2: ABCB1 transporters (WT/E556M/Quadruple) are identically expressed and incorporated within the cell surface, a very important control experiment for the subsequent results. By comparing the conformation of ABCB1 in untreated, CsA-treated and ATP depleted cells the authors concluded that the E556M derivative obtains an ATP-induced trapped state. Since the assay reports on the translocator conformational state, it means very importantly that this conformational event, initiated at the NBDs, is also transmitted to the translocator domains. The trapped state can be completely removed by the Quadruple mutations.

2. Fig.3: The ATP-driven conformations are indeed dependent on ATP concentration, this is very clear from this figure. However, given the fact UIC2-binding is extremely low for the E556M derivative, even in untreated cells (Fig.2B, Untreated, middle column); how did the authors managed to obtain a dose response curve (as a function on MgATP concentration) for the E556M derivative? The authors should introduce a right Y axis in all Fig.3 panels with the absolute together with the relative (left Y axis) values of UIC2 binding.

3. Fig.4: I agree with the fact that in the quadruple mutant the substrate stimulated (which is the one meaningful for function) steady-state ATPase activity is restored, while completely abolished in the E556M derivative.

The steady-state basal ATPase activity of the quadruple mutant is though approximately half of that of wild-type. Is that due to the fact that steady-state ATP hydrolysis occurs only in NBSII?

I would like to insist on the term steady-state ATP hydrolysis instead of just ATP hydrolysis also for issues that I will detail afterwards.

4. Fig.5A: Indeed, the restoration of the substrate stimulated ATPase activity (Fig.4) leads to restoration of transport. The fact that steady state ATPase activity comes likely only from NBSII in the quadruple derivative, this seems to lead to half transport activity. The authors should comment on that. If that is true, this implied that ABC exporters with one degenerate NBS, retain half on the activity with the same (half) ATP consumption (Fig.4).

Fig.5B: In contrast, the results for Calcein-AM are not so clear. Here I would argue a very small difference between E556M and the quadruple derivative, and a huge difference between the quadruple derivative and wild type [especially if one would subtract the minus transporter values (first column)]. The authors should comment on that. Does a degenerate site increase transport selectivity?

5. Fig.6: The results indicate that NBS1 of the quadruple mutant is more “relaxed” than that of WT and the E556M derivative. This according to the authors is the reason of by-passing the ATP trapped state of E556M. This would mean that the dynamics of canonical and degenerate NBS1 are different. This is an important conclusion, but the authors should validate it by examining the dynamics of NBS1 of ABCB11.

6. Fig.7A: The results indicate that E556M restrict the dynamics of the NBD dimer and the quadruple mutant restores back partially such dynamics. This seems to denote that increased dynamics are needed for function. The authors should comment on that.

For me such results are counter-intuitive compared to the results in Fig.6. As the NBS1 is similarly packed is WT and E556M and more relaxed in the quadruple mutant, I would not expect such enhanced dynamics only for WT.

Again, similarly to point5, the NBD/NBD dynamics of ABCB11 should also be examined and compared.

7. Fig.7B: Here the authors examine the correlation between the Qloop1-Signature2 and the Signature1-Signature 2 distance. The results for WT and E556M coincide more than those between the WT and the functional quadruple mutant.

I would expect that ABCB11 would have a similar correlation with the quadruple mutant, indicative of a different mode of action for degenerate NBSs; and for this is should be examined.

8. Fig8: Here the results indicate that the distance between the Q loops are similar for WT and the Quadruple mutant; while the dynamics of the D loops are more similar between WT and the E556M derivative.

From my points 5-8 it becomes apparent that the authors should try to relate the properties of the NBD dimer that are critical for function (thus the same for WT and the quadruple derivative) and uncouple them from those that describe probably different dynamics occurring in NBD dimers harboring degenerate sites.

As mentioned, for this to be accomplished authors should investigate

1. The corresponding dynamics in ABCB11

2. The effect of critical mutations that affect function and presumably dynamics, alike the double mutant Q475A/Q118A the authors mention in line 305 would be very helpful to be investigated.

9. The authors claim multiple times within the paper, that E556M mutation is deficient in ATP hydrolysis: line 374, 67, 91,47 etc, implying that E556 acts as the catalytic carboxylate to cleave the bond between the beta and gamma phosphate. Thus, authors conclude that in the E556M (and consequently in the degenerate NBSs) ATP is not hydrolyzed and either remains bound to keep the NBDs in the close proximity (responsible for the locked, non-functional conformation in the E556M derivative) or ATP unbinds (lines 398-401). This is contrary to previous published data, that indicate that ATP hydrolysis is achieved but when E556 is mutated, defects are encountered in multiple ATP hydrolysis rounds (Biochemistry 2002,41,13989-14000 Sauna et al & JBC, 2004,279, 31212-31220 Tombline et al). The authors did not proved that hydrolysis is abolished and to my knowledge there is zero evidence in the literature that E556 is the catalytic residue in ABCB1.

In line with this, there are controversial results in the literature on the role of the conserved E of Walker B motif (nicely summarized of page 328 of the review Davidson, Chen et al; 2008, doi:10.1128/MMBR.00031-07)

The fact that E556 is not the catalytic residue, seems to be consistent with the findings of the authors, given the fact that it does not contact the beta/gamma phosphate during their simulations (lines 255-263).

The authors should comment on this very important point in the manuscript. According to their simulations, what is their role of E556?.

As the role of that residue -the main player in their study-, is controversial, they should refer extensively in the introduction about the function of the Walker B motif E residue, and also adapt all their discussion statements and conclusions with respect to the correct role of E556.

Minor Issues:

1. For clarity, I strongly believe that authors should keep the orientation of the NBD dimer identical throughout the paper (Fig.1, Fig.6C, Fig.7D,E, Fig.8C/D).

They should create a supplementary figure having multiple orientations of the NBD dimer, indicating with arrows the rotations to go from one to the other. This should only be a ribbon representation in which all residues relevant for this study and important regions (WalkerA,B; signature motifs, etc) are indicated.

2. Authors claim that the difference between the NBD dimeric interface between ABCB1 and ABCB11 is limited to 4 residues (line 126).

Authors should provide a supplementary table in which all interface interactions in the 2 proteins are indicated

3. The alignments to conclude that the 4 differing residues between ABCB11 and ABCB1 are conserved in ABCB11 should be placed in the Supplement

Reviewer #2: In this manuscript, Goda et al use various experimental and computation techniques to show that while mutating E556 in one of the nucleotide binding sites abolishes both the catalytic and transport activities of ABCB1 (as previously known), employing three more mutations in the NBD interface based on the ABCB11 sequence will result in the gain of both catalytic and transport activities (at least for some substrates). The experiments and arguments are generally straightforward and the manuscript reports very valuable results; however, there are some issues that need to be addressed.

Major points:

1) Convergence of simulation data: The computational side of this manuscript could have played a very important role in elucidating the important experimental observations made. However, there is a major issue with the simulation data. The simulations are only 400 ns long and although it might be computationally difficult to perform longer simulations, it is difficult to trust the data if a convergence has not been achieved. Some of the time series provided question the reach of convergence. For instance, the bottom panels of Fig. 6B and Fig. 8B show some important events happening towards the end of the simulation. It is very reasonable to assume that if one continues these simulations for another 400 ns or so we may observe significant shifts in various distributions shown in different plots. It is thus important to continue these simulations until some relative convergence is reached.

2) Transport activity of the quadruple mutant: The authors state that (lines 238-239) the quadruple mutant was able to limit Hoechst 33342 and calcein accumulation, though not as efficiently as wild-type ABCB1. However, Fig. 5 shows that the calcein transport activity of the quadruple mutant is much closer to that of the E556M mutant than the wild-type ABCB1. I think it is more accurate to state that the gain of transport function is substrate dependent. It is possible that a higher concentration of calcein could result in a better distinction between the transport activity of the single and quadruple mutant but we have not been provided by such data.

Minor point:

In the Discussion section, it is mentioned that ATP hydrolysis by the NBDs converts the chemical energy stored in the phosphate bonds of ATP into NBD motions. This statement somewhat implies it is the heat released from the hydrolysis that results in the NBD dissociation but even in the absence of this heat, e.g., when the ATPs are not present at all, one expects the equilibrium to shift towards the dissociated NBDs. It is the free energy rather than the heat/enthalpy that matters here.

**Have all data underlying the figures and results presented in the manuscript been provided?**

Reviewer #1: Yes

Reviewer #2: Yes

PLOS authors have the option to publish the peer review history of their article (what does this mean?). If published, this will include your full peer review and any attached files.

Reviewer #1: No

Reviewer #2: No

---

## [Decision Letter · Decision Letter 1]

29 Jul 2020

Dear Dr Stockner,

We are pleased to inform you that your manuscript entitled "Human ABCB1 with an ABCB11-like degenerate nucleotide binding site maintains transport activity by avoiding nucleotide occlusion" has been editorially accepted for publication in PLOS Genetics. Congratulations!

Yours sincerely,

Thorben Cordes, PhD

Guest Editor

PLOS Genetics

Gregory Barsh

Editor-in-Chief

PLOS Genetics

Comments from the reviewers (if applicable):

Based on the feedback from the referees, we would be happy accept the paper provided that the authors add the discussion points by referee #1 and the data are deposited as stated in the manuscript and requested by referee #2.

Reviewer's Responses to Questions

**Comments to the Authors:**

Reviewer #1: In this revised version Stockner and coworkers addressed adequately my comments and concerns.

The only critical point not addressed is represented by the missing simulations I proposed on ABCB11. This experiment is important to demonstrate that the finding of this study (that an occluded state can be “by-passed” by altering the conformational dynamics and not uniquely via ATP hydrolysis) is not limited to an artificially degenerate NBS (ABCB1 E556M), but represents the solution of evolution for all (or many) naturally occurring degenerate NBSs (ABCB11 and others). To my view this is a possibility, as ABCB1 E556M was rendered functional based on ABCB11. Of course, TMDs modulate NBD dynamics, however ABCB1 E556M was rendered functional only by considering the NBDs of ABCB11 and not the entire transporter.

However, I must admit that the authors do not make generic statements with respect to their findings, thus to my view the manuscript is absolutely consistent.

I hope the authors address this generality issue in their subsequent manuscripts and should be done in the Discussion of the current one.

Reviewer #2: The authors have addressed my concerns. However, I have a concern regarding the data availability as described below.

**Have all data underlying the figures and results presented in the manuscript been provided?**

Reviewer #1: Yes

Reviewer #2: **No: **The author states that "all data are fully available without restriction"; however, it is not discussed where the data sets have been deposited. They are certainly not in the supporting information.

PLOS authors have the option to publish the peer review history of their article (what does this mean?). If published, this will include your full peer review and any attached files.

Reviewer #1: No

Reviewer #2: No

**Data Deposition**

http://datadryad.org/submit?journalID=pgenetics&manu=PGENETICS-D-19-01402R1

**Press Queries**

---

## [Editor Report · Acceptance letter]

23 Sep 2020

PGENETICS-D-19-01402R1 

Human ABCB1 with an ABCB11-like degenerate nucleotide binding site maintains transport activity by avoiding nucleotide occlusion 

Dear Dr Stockner, 

We are pleased to inform you that your manuscript entitled "Human ABCB1 with an ABCB11-like degenerate nucleotide binding site maintains transport activity by avoiding nucleotide occlusion" has been formally accepted for publication in PLOS Genetics! Your manuscript is now with our production department and you will be notified of the publication date in due course.

With kind regards,

Matt Lyles

PLOS Genetics

On behalf of:
